# A New Phylogenetic Inference Based on Genetic Attribute Reduction for Morphological Data

**DOI:** 10.3390/e21030313

**Published:** 2019-03-22

**Authors:** Jun Feng, Zeyun Liu, Hongwei Feng, Richard F. E. Sutcliffe, Jianni Liu, Jian Han

**Affiliations:** 1Department of Information Science and Technology, Northwest University, Xi’an 710127, China; 2Early Life Institute, State Key Laboratory of Continental Dynamics, Department of Geology, Northwest University, Xi’an 710069, China

**Keywords:** attribute reduction, information entropy, morphological analysis, phylogenetic tree

## Abstract

To address the instability of phylogenetic trees in morphological datasets caused by missing values, we present a phylogenetic inference method based on a concept decision tree (CDT) in conjunction with attribute reduction. *First*, a reliable initial phylogenetic seed tree is created using a few species with relatively complete morphological information by using biologists’ prior knowledge or by applying existing tools such as MrBayes. *Second*, using a top-down data processing approach, we construct concept-sample templates by performing attribute reduction at each node in the initial phylogenetic seed tree. In this way, each node is turned into a decision point with multiple concept-sample templates, providing decision-making functions for grafting. *Third*, we apply a novel matching algorithm to evaluate the degree of similarity between the species’ attributes and their concept-sample templates and to determine the location of the species in the initial phylogenetic seed tree. In this manner, the phylogenetic tree is established step by step. We apply our algorithm to several datasets and compare it with the maximum parsimony, maximum likelihood, and Bayesian inference methods using the two evaluation criteria of accuracy and stability. The experimental results indicate that as the proportion of missing data increases, the accuracy of the CDT method remains at 86.5%, outperforming all other methods and producing a reliable phylogenetic tree.

## 1. Introduction

In biology, phylogenetic inference is an important research focus with the goal to discover the evolutionary history of species and their relationships. The goal of phylogenetic inference is to assemble a tree representing a hypothesis of the evolutionary ancestry of a set of genes, species, or other taxa. Figure 1A–H shows selected fossil images, a species-against-attributes matrix, and the results of the phylogenetic inference analysis.

Due to incomplete records, there are almost always missing values for fossils; for example, the parts marked with “?” in image (C) in Figure 1 indicate missing data. Thus, it may be difficult to support the results of the phylogenetic inference analysis under such circumstances. To solve this problem, four methods have been developed to deal with missing data. First, a certain proportion of incomplete species or attributes can be removed [2,3]. However, in many cases, the exclusion of incomplete species and attributes is carried out in an arbitrary manner without specific explanations or reasons [3,4,5,6]. Second, the number of attributes can be increased [7,8]. Research results showed that if the overall number of attributes in the analysis was sufficiently large (more than 1000 attributes), a phylogenetic inference method accurately reconstructed the position of highly incomplete taxa (e.g., 95% missing data) [2,3]. However, due to the simple structure of early paleontology, species often have less than 200 attributes. When the absence rate is high, a common phylogenetic inference cannot be accurately inferred. Highly incomplete taxa may produce multiple equally parsimonious trees and poorly resolved consensus trees, resulting in low phylogenetic accuracy [9]. Third, the missing values can be filled in. In the Hennig86 [10] and PAUP [11] programs, for example, each unspecified attribute is randomly assigned a value that is suitable for the attribute. Each of these three methods of dealing with missing data has its strengths and weaknesses, but none reflect the true value of the missing data. Fourth, The species-against-feature matrix with missing data can transform into a suitable sparse expression form by a sparse sampling algorithm, and the reconstruction algorithm is used to reconstruct the sampling point. Sparse signal recovery theory shows that this method can accurately reconstruct data. A common sparse representation method is wavelet analysis [12,13,14,15]. By sparse representation of the data, wavelet analysis could potentially be applied to the task of recovering missing phylogenetic information. It has been successfully applied to signals, images, gene classification and so on [16,17]. However, in the study of morphological phylogenetic analysis, it is a method that is little studied but worth trying.

In addition, there are three main approaches based on the principle of optimality for inferring the phylogenetic tree, namely maximum parsimony (MP) [18], maximum likelihood (ML) [19] and Bayesian inference (BI) methods [9,20]. The ML and Bayesian methods are commonly used probabilistic approaches based on matrices containing only gene data from living species [21]. However, since DNA is usually not available for fossil taxa, only the fossil occurrence dates are used in time-calibrated phylogenies [22]. Moreover, researchers have found that the ML and Bayesian methods do not deal effectively with missing morphological data [20]. MP is well known to be non-deterministic polynomial-time (NP)-hard [23]. Given the large number of taxonomic groups, the only effective method of obtaining the optimal phylogenetic tree is to perform a heuristic search. However, studies have shown that MP may fall into a local optimum. Therefore, complex and flexible heuristics are needed to ensure that the tree space is fully explored.

Our motivation is to introduce a phylogenetic inference method that reduces the impact of missing data. In this paper, we propose an evolution analysis algorithm based on bi-directional cognitive processing; we call this approach phylogenetic deduction based on a concept decision tree (CDT). We use a cognitive model to reduce the search scope caused by incomplete data. In this model, a priori knowledge of relatively complete species is used to create a highly reliable phylogenetic tree as an initial seed. Attribute reduction [24] based on rough sets [25] is used to construct multiple concept-sample templates for each node of the initial seed tree by removing unrelated or unimportant attributes in order to improve the classification or decision-making [26], thereby reducing the impact of missing data. We apply a matching algorithm to evaluate the matching degree between species’ attributes and the nodes’ concept-sample templates; hence we determine the location of the species by a serial search in the phylogenetic tree. Therefore, the global combinatorial explosion problem is decomposed into a classification framework that prevents instability. Compared with the traditional parallel phylogenetic inference process applied to all species, our method greatly reduces the computational scale and complexity of the task. Gradually, a complete phylogenetic tree is established.

Here we compare our method with the MP, ML, and BI methods using morphological datasets with different amounts of missing data. We show that the proposed algorithm makes a contribution to the field because it enables the construction of morphological data with an accuracy of 86.5% whereas the MP, ML, and BI methods provide accuracies of 85.5%, 82.8%, and 85.1%, respectively. We also compare the stability of the methods to establish the tree. The experimental results show that the variance of our method and the other methods is 0.0872. Therefore, a stable phylogenetic tree can be constructed.

The rest of the paper is organized as follows. Section 2 introduces the framework of the CDT algorithm. The process of developing concept-sample templates based on genetic algorithms (GAs) is described in Section 3. Section 4 presents the experimental results of the CDT and the discussion. Finally, Section 5 provides the conclusions of the study.

## 2. Framework of the CDT Algorithm

The objective of the CDT algorithm is to construct a phylogenetic tree *T* for a set of species *S*, expressed as T=(V,E) where V←S. We input a species-against-attribute matrix SOA for a set of species *S*. The species are sorted in order of completeness from high to low, which is denoted as $S=\left\{s_{1},s_{2},\ldots ,s_{n}\right\}$. For each species $s_{j}\;\left(1\leq j\leq n\right)$, there are *m* attributes, which are defined as $A=\left\{a_{1},a_{2},\ldots ,a_{m}\right\}$. We divide *S* into $sub_{1}$ and $sub_{2}$, where $sub_{1}=\left\{s_{1},s_{2},\ldots ,s_{i}\right\}$ and $sub_{2}=\left\{s_{i+1},s_{i+2},\ldots ,s_{n}\right\}$. The species in $sub_{1}$ are relatively complete, whereas those in $sub_{2}$ are missing many attributes.

The framework of the phylogenetic inference based on the CDT is shown in Figure 2.

We divide the framework into four steps as follows:

(1) *The establishment of the initial seed tree*

Due to the ambiguity of phylogenetic tree construction, the initial concept establishment is very important because it reduces the complexity of the subsequent steps. During the analysis of species evolution, we first apply either biologists’ prior knowledge or common software tools (such as MrBayes [27], PAUP* [11], or TNT [28]) to a set of relatively complete species $sub_{1}=\left\{s_{1},s_{2},\ldots ,s_{i}\right\}$ in order to build a reliable phylogenetic tree $T_{init}$ as an initial seed, where $T_{init}=\left(V^{*},E^{*}\right)$, $V^{*}=sub_{1}$.

(2) *The generation of decision points in the initial seed tree*

To take advantage of the established concepts, we perform attribute reduction on the rough set at each branch node of the initial seed tree $T_{init}$ by analyzing the species $sub_{1}$’ location. In this way, we obtain the concept-sample templates for the branch nodes in $T_{init}$. Therefore, the branch nodes have decision-making functions that become decision points. Correspondingly, the phylogenetic seed tree becomes the decision tree $T_{deci}$, which provides the basis for the grafting of species with missing data.

(3) *Species grafting*

For species $s_{j}$ in $sub_{2}$, we can determine its location in the phylogenetic tree by matching the species’ attributes with multiple concept-sample templates of each decision point in a top-down manner.

(4) *The construction of a complete phylogenetic tree*

The evolutionary process starts with the most reliable species $s_{i+1}$ in $sub_{2}$, followed by grafting it onto the tree, as described in Step 3. The next species $s_{i+2}$ is then added, and so on, finishing with species $s_{n}$. In this way, a complete phylogenetic tree $T_{final}$ is constructed.

In this paper, we focus on the generation of decision points in the initial seed tree (Section 3) and species grafting (Section 4).

## 3. Construction of Multiple Concept-Sample Templates

The internal nodes in the phylogenetic tree are an important decision-making basis for phylogenetic inference. Therefore, we transform the internal nodes into decision points. Due to a large number of missing and inconsistent attributes, traditional pattern recognition methods are not applicable. Therefore, a method is required to provide decision-making attribute sets for the internal nodes.

We propose to generate multiple concept-sample templates for the internal nodes based on the species’ location. The purpose of rough set attribute reduction is to remove unrelated or unimportant attributes in order to improve classification or decision-making [21,29]. Attribute reduction has been shown to be an NP-hard problem for combinatorial optimization [22,23]. However, in many applications, it is necessary to find only one minimum attribute reduction. On the other hand, because morphological data in Paleontology are often missing many values, we need to use multiple concept-sample templates to make full use of the data. In this study, we use entropy-based genetic algorithms (GAs) [24] to find the optimal template sets heuristically because they can simulate the optimal solution of a natural evolutionary process, and phylogenetic inference is essentially part of the study of evolution.

### 3.1. The Design of the Genetic Algorithm for Attribute Reduction

In this section, we introduce the details of the GA to deal with attribute reduction in the rough set theory.

#### 3.1.1. Encoding Method

A variable-length decimal array of one-dimensional strings represents the chromosome. The length of the chromosome equals the number of the species’ attributes, i.e., *N*. Each gene bit corresponds to an attribute in the chromosome. Each gene bit in the chromosome is numbered 1−N, and the corresponding code ranges from 0 to the number of the species’ attributes, where 0 denotes that the attribute is not selected and *i*
(0<i<N) denotes that the *i*th attribute is selected as the attribute of the concept-sample template. The chromosomes in the initial population are generated using uniformly distributed random numbers.

When the length of the chromosome is *N*, each chromosome corresponds to a unique set of concept-sample templates for a total of ${\left(N+1\right)}^{N}$, as shown in Table 1 below:

For example, Table 2 shows the encoding method of a chromosome with N=10. Sites 8 and 9 have the same value 9, indicating that attributes 8 and 9 belong to the same concept-sample template. Site 10 has value 8 and the other sites have different codes; therefore, attribute 10 represents a single concept-sample template. For example, if the template set X for a decision point is $\left\{1\;3\;0\;4\;5\;7\;10\;9\;9\;8\right\}$, it contains $\left\{1\right\}\left\{2\right\}\left\{4\right\}\left\{5\right\}\left\{6\right\}\left\{7\right\}\left\{8,9\right\}\left\{10\right\}$.

#### 3.1.2. Fitness Function

The fitness of a chromosome determines the probability with which it will be inherited by the next generation. Here, the fitness of a chromosome is calculated by reference to the concept-sample template set generated by it. According to the principle of attribute reduction, *B* represents the attribute subset of the present mapping, $C=\left\{c_{1},c_{2},\ldots ,c_{r}\right\}$ represents the attribute set of the species, and $D=\left\{0,1\right\}$ represents the class label of the species belonging to the node.

**Definition** **1.**
*Let $U=\left\{x_{1},x_{2},\ldots ,x_{n}\right\}$ be a non-empty finite set of objects, called the domain. X⊆U, X≠⊘ the B-lower approximation set of X is defined as follows:*
(1)$$\underline{B}\left(X\right)=\left\{x\in U|{\left[x\right]}_{R}\subseteq X\right\}$$
*where ${\left[x\right]}_{R}$ denotes an equivalence class determined by object x.*


**Definition** **2.**
*Assuming that C,D⊆A, X∈U/D, the lower approximation set is defined as follows:*
(2)$$POS_{C}\left(D\right)=\underset{x\in U/D}{\cup }\underline{B}\left(X\right)$$
*That is, the lower approximation set is obtained from all of the sets contained in X.*


If $POS_{B}\left(D\right)=POS_{C}\left(D\right)$, we calculate $\frac{\left|C\right|-r_{n}}{\left|C\right|}$ and substitute it into the fitness function of Equation (Equation 3). If $POS_{B}\left(D\right)\neq POS_{C}\left(D\right)$, $\frac{\left|C\right|-r_{n}}{\left|C\right|}=0$. The fitness function is defined as follows:(3)$$F=\sum_{n=1}^{L}\frac{\left|C\right|-r_{n}}{\left|C\right|}$$
where *L* represents the number of concept template sets in the chromosome, $\left|C\right|$ represents the number of species attributes, *n* represents the *n*th concept-sample template, and $r_{n}$ represents the number of attributes in the *n*th template.

#### 3.1.3. Selection Operator

We use the roulette wheel selection method to choose the best individual to continue to the next generation. Individuals are selected with a probability proportional to their fitness values [30]. If a population $G=\left\{X_{1},X_{2},\ldots ,X_{pop\_size}\right\}$ (pop_size is the population size) and the fitness of the individual $X_{i}\in G$ is $F(X_{i})$, the probability of an individual $X_{i}$ being selected is $P_{i}$:(4)$$P_{i}=\frac{F(X_{i})}{\sum_{j=1}^{pop\_size}F\left(X_{j}\right)}$$
$P_{i}$ reflects the proportion of the fitness value of the individual $X_{i}$ with respect to the sum of fitness values of all individuals.

In order to ensure that the best individuals survive to the next generation, we use the optimal preservation strategy [31]. If the fitness value of the worst individual in the current generation is less than the fitness value of the best individual in the previous generation, we use the best individual in the previous generation to replace the worst individual in the current generation. In the case of more than one optimal individual, the optimal individual is randomly selected to replace the worst individual.

#### 3.1.4. Crossover Operator

The crossover operation uses a random single-point crossover strategy. An individual is chosen to take part in the crossover at a certain probability $P_{c}$. All selected individuals are randomly paired. For each pair of individuals, a cross-point is selected randomly. Some of the chromosomes of the paired individuals are exchanged at the cross-point. In this way, the next generation of individuals is generated.

#### 3.1.5. Mutation Operator

The mutation operations use the “basic bit” variation. For each chromosome selected with probability $P_{m}$, its mutation point is specified by a random probability and the value at the specified mutation point becomes another state value. In this way, we can generate further members of the next generation to improve the performance of the heuristic search.

#### 3.1.6. Modification Operator

**Step 1:** Calculate the mutual information I(C;D) of the condition attribute set *C* and the decision attribute set *D*. The mutual information [32] of *C* and *D* is defined as
(5)I(C;D)=H(D)−H(D|C)
where $H\left(X\right)=-\sum_{i=1}^{n}P\left(x_{i}\right)\log_{b}\left(x_{i}\right)$ and the conditional entropy of *X* and *Y* is defined as $H\left(X|Y\right)=-\sum_{i,j}p\left(x_{i},y_{j}\right)\log\frac{p(x_{i},y_{j})}{p(y_{j})}$. When *X* and *Y* are independent, I(X;Y)=0; otherwise, this index is positive [32,33] and it increases with the degree of dependence between the components $x_{i}$ and $y_{i}$.

**Step 2:** Calculate I(Reduct;D) and I(C;D). If I(Reduct;D)<I(C;D) then repeat steps 3 and 4; otherwise, end the modification;

**Step 3:** Select attribute *a* in C−Reduct so that $SGF\left(a,Reduct,D\right)=H\left(D|Reduct\right)-H\left(D|Reduct\cup \left\{a\right\}\right)$ reaches the maximum value. SGF(a,Reduct,D) reflects the increment of mutual information when *a* is added to Reduct. According to the definition of attribute importance of the mutual information, we select the attribute and set it to $a_{j}$;

**Step 4:** Change the bit corresponding to $a_{j}$ from 0 to *j* and return to step 2;

### 3.2. Algorithm Description

**Input:** An attribute table of Species *C*, the class label of the species *D*

**Output:** Concept-sample template sets $Reduct_{i}\left(i=1,2,\ldots ,n\right)$ for each internal node

**Step 0:** Set the parameters: chromosome size *m*, population size pop_size, crossover probability $P_{r}$, mutation probability $P_{m}$, and maximum generation maxgen. Let generation gen=0.

**Step 1:** Generate pop_size chromosomes randomly.

**Step 2:** Calculate the fitness value of each chromosome.

**Step 3:** Perform crossover on individuals selected with probability $P_{r}$.

**Step 4:** Perform mutation on individuals selected with probability $P_{m}$.

**Step 5:** Create the new population. Select pop_size individuals from the parents and offspring for the next generation by the roulette wheel selection method.

**Step 6:** Perform modification of the individuals.

**Step 7:** Stop calculating. If gen=maxgen, then output the corresponding concept-template collection $Reduct_{i}\left(i=1,2,\ldots ,n\right)$ and stop, else let gen=gen+1 and return to Step 2.

## 4. Species Grafting Algorithm (SGA)

### 4.1. Description of SGA

In the phylogenetic seed tree, the concept-sample template sets for each decision point provide a basis for grafting the species. We calculate the matching degree of the species’ attributes and each node’s concept-sample template sets in the phylogenetic seed tree in a top-down manner. In this way, we can identify the location of each species in the phylogenetic seed tree and gradually complete the grafting process.

The process of species grafting at each decision point is shown in Figure 3. As shown in the tree branching, the species are divided into *A* and *B* subtrees. *Q* represents the attribute of the grafted species. *L* indicates the attribute values of the sample templates of the *A* subtree. *R* indicates the attribute values of the sample templates of the *B* subtree. Let *K* be the number of concept-sample templates for the decision point in the concept decision seed tree. Suppose *m* is the number of concept-sample templates that match the *A* subtree and *n* is the number of concept-sample templates that match the *B* subtree; in this case *m* and *n* are initialized to 0. For each of the attribute values L,R, if $L_{i}\subseteq Q$ (or $R_{i}\subseteq Q$), that is i.e., the species’ attribute *Q* contains the attribute values $L_{i},R_{i}$ for each decision point, we can determine that *Q* belongs to the *A* or *B* subtrees and let m=m+1 (or n=n+1). If it neither belongs to the *A* subtree nor the *B* subtree, or it cannot be assigned because *Q* contains missing values, then *m* and *n* are not accumulated.

When the species’ attribute *Q* is assigned to subtree A or subtree B of the root decision points, we continue to perform top-down matching using the next decision point of the subtree. Using these steps, the species’ location in the tree is determined. Finally, the species are grafted into the ultimate decision point of the phylogenetic tree.

In the grafted species, the proportion of missing values differs for each species. In order to obtain a stable phylogenetic tree, the grafting is conducted one-by-one taking into account the integrity of the species’ attributes. When all species have been grafted, a complete phylogenetic tree has been constructed.

### 4.2. Detailed Example of SGA

In this section, an example is given to illustrate the specific implementation of the SGA. As shown in Figure 4(1), an initial phylogenetic seed tree is constructed based on the species-against-attributes matrix. Then, we use the method described in Section 3 to create multiple concept-sample templates for each internal node in the tree, as shown in Figure 4(2). We consider the internal nodes *R*, $N_{1}$, $N_{2}$ in order from top to bottom. From the phylogenetic seed tree in stage (2) of the diagram, the *R* node divides the species into two groups: the left subtree (species *X*, *Y*, *Z*) and the right subtree (species *I*). The concept-sample templates of the *R*-node after attribute reduction are $\left\{1\right\}$, $\left\{3,6\right\}$, $\left\{4,8\right\}$, which means that these templates also correctly divide the species. For example, attribute 1 divides the left subtree (*X*, *Y*, *Z*) and right subtree (*I*); we also know from the species-against-attributes matrix that the corresponding left subtree has a value of 1 or 2, which is recorded as the attribute value set $L_{1}$ of the concept sample template. The corresponding right subtree has a value of 0, which is recorded as the attribute value set $R_{1}$ of the concept sample template in Figure 4(3). Similarly, we can determine the attribute value set for the other sample templates.

The grafted species *G* is matched with the concept-sample templates of the decision points to determine the grafting position. The *G*′ attributes are compared with the values of the node *R*’s left subtree and right subtree. In (3), *m* and *n* are initialized to 0. For the first collection of attribute values ($L_{1}$, $R_{1}$), since $a_{1}=1$ in species *G* and $a_{1}\in L_{1}$, it conforms to the left subtree Tree *A*; therefore, m=m+1 provides the value 1. For the second collection of attribute values ($L_{2}$, $R_{2}$), because $a_{3}=2$, $a_{6}=0$, species *G* corresponds to the left subtree Tree *A*; therefore, m=m+1 results in 2. For the third ($L_{3}$, $R_{3}$), because in the species *G*, $a_{4}=1$, $a_{8}$ represents missing data; therefore *m* and *n* remain unchanged. At this point, the attribute value set traversal of node *R* is completed. Since m>n, the left subtree *A* is selected. For node ($N_{1}$, $N_{2}$), a similar operation is performed from the top down as shown in Figure 4(4,5). Finally, the position of the species *G* grafting is determined, as shown in Figure 4(6).

During the grafting of multiple species, it may happen that some species cannot be assigned and grafted at the same decision point, resulting in the phenomenon of ‘species-stacking’, i.e., the creation of a polymorphic tree. In such a case, the biologists cannot determine the interspecies relationship between the species through the phylogenetic tree, which affects the accuracy of the evolutionary relationship. In this study, we use the Wagner formula [34] to adjust the structure of the tree by calculating the difference between species:(6)$$d\left(A,B\right)=\sum_{i=1}^{t}\left|X\left(A,i\right)-X\left(B,i\right)\right|$$
where $d\left(A,B\right)$ is the difference between species A,B; *t* is the number of attributes; $X\left(A,i\right)$ is the state of attribute *i* for species *A*; $X\left(B,i\right)$ is the state of attribute *i* for species *B*.

As shown in Figure 5, if species A,B, and *C* are unions, $d\left(A,B\right),d\left(A,C\right),$ and $d\left(B,C\right)$ are calculated. If $d\left(A,B\right)<d\left(B,C\right)$, then *A* and *B* are closer and we merge *A* and *B*. If $d\left(A,B\right)>d\left(B,C\right)$, then *B* and *C* are closer and we merge *B* and *C*, etc. We thereby minimize the generation of polymorphic trees.

## 5. Experimental Results

To assess the accuracy and reliability of the CDT, we conducted experiments on six species datasets. The summary information for the datasets is shown in Table 3.

The datasets were used to construct phylogenetic trees using our CDT algorithm as well as three other standard methods, namely MP, ML, and BI. The specific steps are described in Section 5.1. The grafting results of CDT were compared to the accepted tree topologies (model trees) that are part of the datasets. The results were then compared.

### 5.1. CDT Accuracy Analysis

The accuracy rate of the assignment of a species, i.e., the accuracy of the species’ phylogenetic analysis, depends on the node path of that species. The path of a species in a phylogenetic tree model accepted by biologists is considered to be the standard path sequence $Seq_{s}$. The path sequences of the grafted species $Seq_{c}$ obtained from the CDT, MP, ML, and BI methods were compared with the standard path sequences. $Seq_{s}\cap Seq_{c}$ denotes that $Seq_{c}$ matches the standard sequence $Seq_{s}$. $\left|Seq_{s}\cap Seq_{c}\right|$ is the number of path matching species and $\left|Seq_{s}\right|$ is the total number of standard sequence species. The accuracy can be expressed by Equation (Equation 7). For example, if $Seq_{s}=\left\{1,2,4,5,8,10\right\}$ and $Seq_{c}=\left\{1,2,4,5,8,9\right\}$, then $acc=\frac{5}{6}\approx 83.3\%$.
(7)$$acc=\frac{\left|Seq_{s}\cap Seq_{c}\right|}{Seq_{s}}\times 100\%$$

To verify the performance of the CDT algorithm, the attributes of the species which are to be grafted are randomly chosen to be incomplete. The missing proportions are 0%, 10%, 20%, 30%, 40%, 50%, 60% and 70%. On the basis of different proportions of missing data, we apply the CDT algorithm for species grafting and the MP, ML, and BI methods to establish phylogenetic trees. The bootstrap method [41,42] is used to resample the data set 1000 times and the average accuracy of the four methods is calculated. For six species datasets in Table 3, the accuracies of the four methods of phylogenetic analysis under different proportions of missing data are shown in Figure 6.

We observe the following:

$\left(1\right)$ In general, an increase in missing data results in insufficient information and a decrease in accuracy.

$\left(2\right)$ When the proportion of missing data is less than 10%, the accuracies are similar for the different methods, i.e., the species can be classified accurately.

$\left(3\right)$ The proposed method significantly improves the accuracy of the results, especially for datasets with many missing data (missing proportions > 40%). This occurs because as the species’ number of attributes increases, the amount of data used for the concept-sample templates increases; although the proportion of missing data increases, it is much easier to assign the species to the correct location.

The average accuracies of the CDT, MP, ML and BI methods are shown in Table 4. The accuracy of the CDT method was 86.5% whereas the accuracies of the MP, ML, and BI methods were 85.5%, 82.8%, and 85.1%, respectively, indicating that the proposed method had the highest average accuracy.

### 5.2. CDT Reliability Analysis

To evaluate the reliability of the CDT method, we used the tree length [43] to determine the optimality criteria. In phylogeny, the length of the phylogenetic tree is a parameter for evaluating morphological changes in the tree, i.e., the number of changes in the attributes. The shorter the tree length, the more reliable the phylogenetic tree is. Therefore, a phylogenetic tree with the lowest number of changes in the attribute state is preferred.

We used the results of the phylogenetic tree described in Section 5.1 and calculated the tree length separately as shown in Figure 7.

In Figure 7, it is observed that grafted species with different proportions of missing data have little effect on the tree length. The data in Table 5 were obtained by analyzing the results in Figure 7.

Table 5 shows that the tree length of the phylogenetic tree is similar for all four methods and the average variance of the tree length is 0.0872. Therefore, our method is as reliable as the other methods.

### 5.3. Phylogenetic Inference on Cambrian Lobopodians

In this study, we apply the CDT to the phylogenetic analysis of the Cambrian lobopodians. The Cambrian lobopodians paleontological morphological dataset [1] contains large amounts of missing data; for example, the species *Opabinia* has 32% missing data, while *Hadranaxa* and *Orstenotubulus* have 48% missing data. The species *Opabinia*, *Hadranaxa*, and *Orstenotubulus* were sequentially used for grafting to construct phylogenetic trees, as shown in Figure 8. The results show that our method provides a phylogenetic tree that is consistent with the assessment of paleontologists.

## 6. Conclusions

In this paper, we used bi-directional cognitive and concept-driven processing for the process of phylogenetic inference. Using prior knowledge of phylogenetic analysis, we generated a phylogenetic seed tree and used genetic attribute reduction to construct concept-sample template sets for each decision point using a top-down algorithm. Subsequently, top-down template matching was used to determine the grafting position of the species containing missing values in the phylogenetic seed tree. The experimental results show that the CDT method had high accuracy and stability and resulted in a phylogenetic tree that was familiar to biologists. The proposed method solves the problem of creating a stable phylogenetic tree when much of the data are missing.

## Figures and Tables

**Figure 1 entropy-21-00313-f001:**
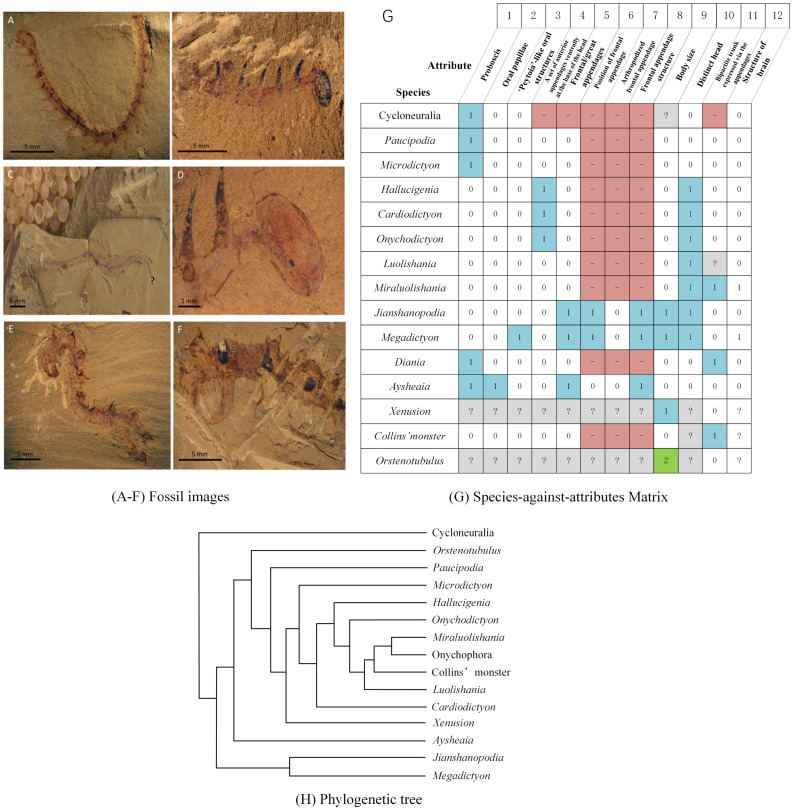
An example of phylogenetic inference. Photographs (**A**–**F**) show examples of Cambrian Chengjiang Lagerstätte fossils. (**G**) is a morphological attribute matrix, where the rows represent species and the columns represent attributes. In the column labels of the matrix, the first row represents the attribute number and the second row corresponds to the attribute name. (**H**) is a phylogenetic tree for selected lobopodians and arthropods from the early Cambrian era [1].

**Figure 2 entropy-21-00313-f002:**
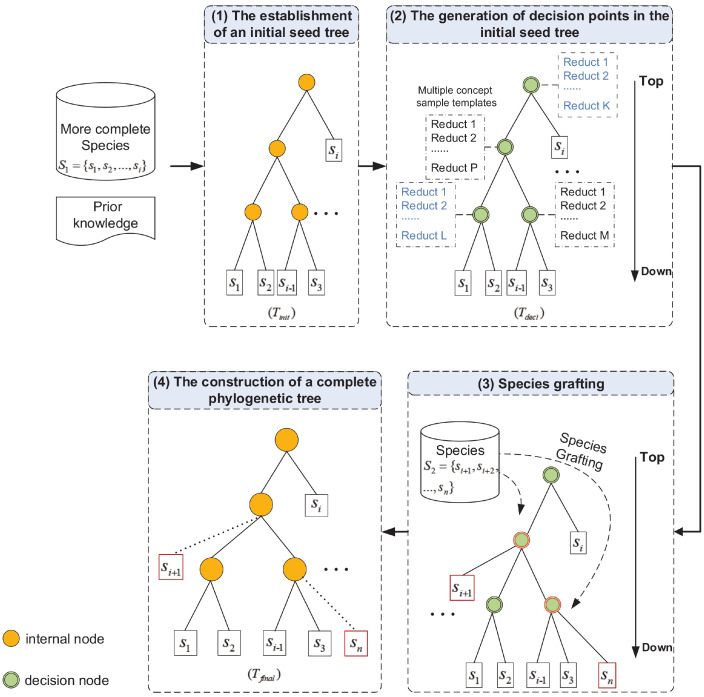
The framework of phylogenetic inference based on the Concept Decision Tree.

**Figure 3 entropy-21-00313-f003:**
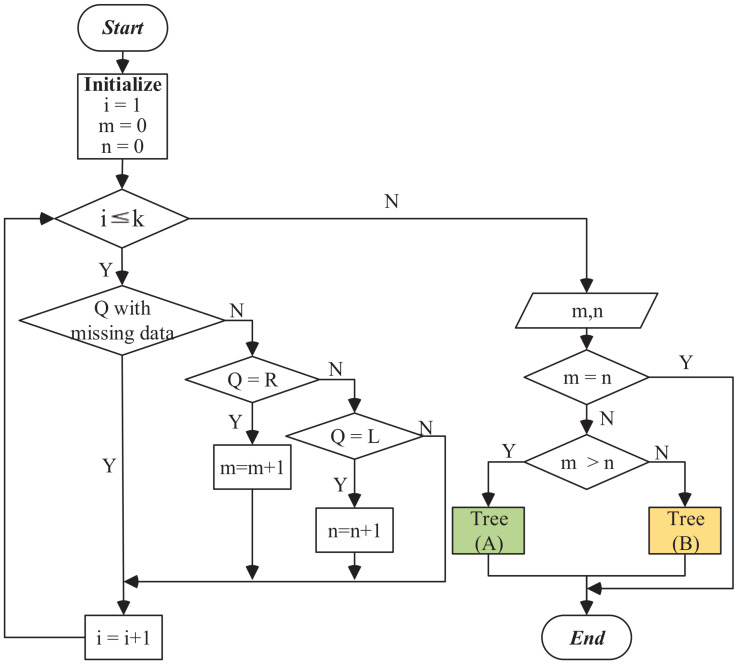
The strategy of species grafting in a single decision node.

**Figure 4 entropy-21-00313-f004:**
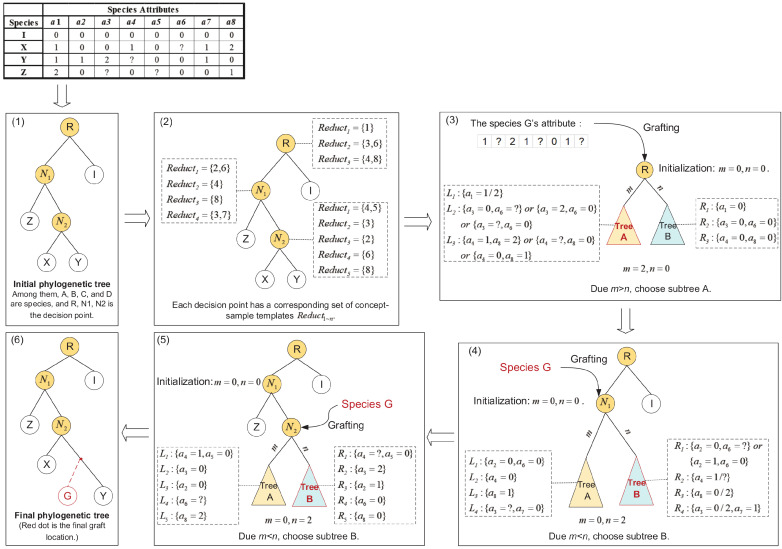
An example of the species grafting algorithm. The red dot indicates the final graft position of the species *G*.

**Figure 5 entropy-21-00313-f005:**
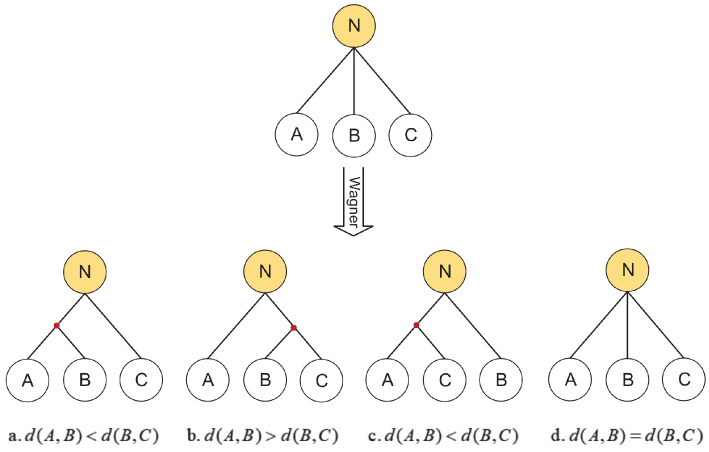
An example of handling polymorphic trees.

**Figure 6 entropy-21-00313-f006:**
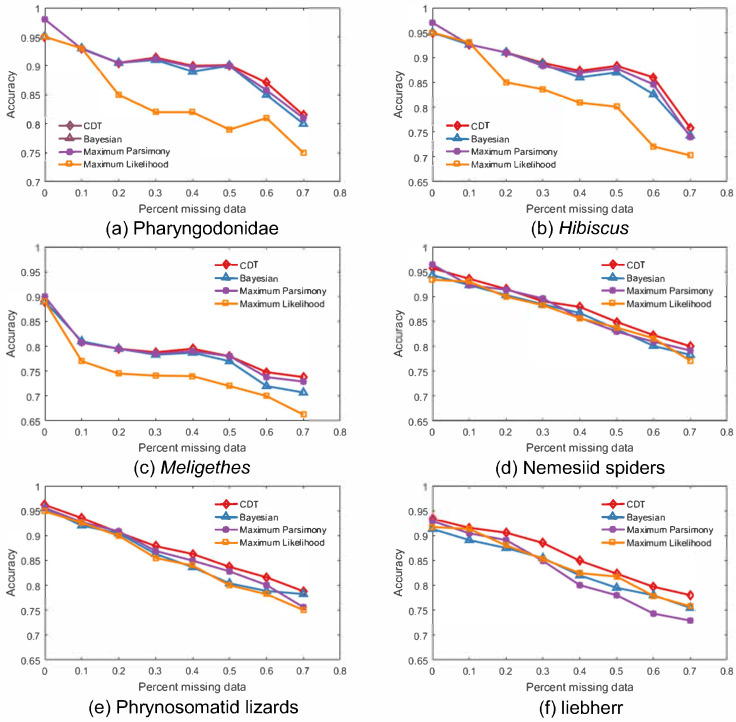
Accuracies of phylogenetic analysis for different proportions of missing data.

**Figure 7 entropy-21-00313-f007:**
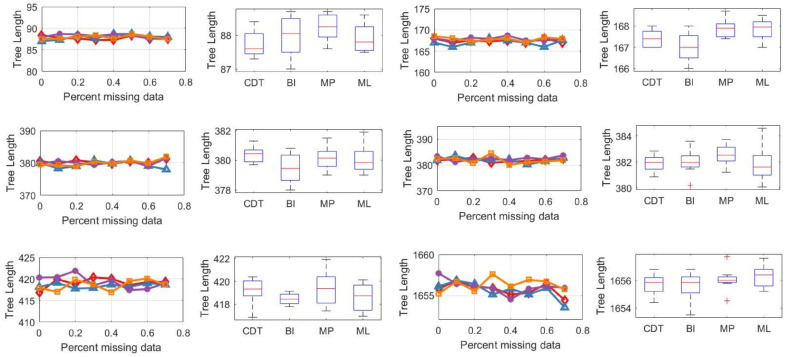
The tree length for different proportions of missing data and for different methods.

**Figure 8 entropy-21-00313-f008:**
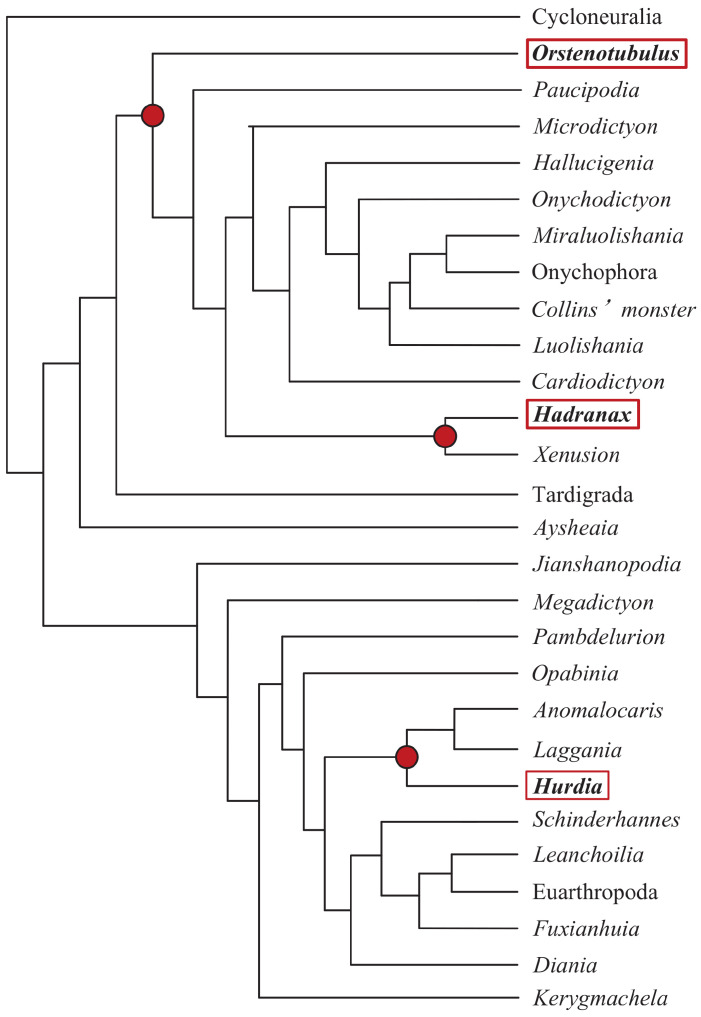
A paleontological phylogenetic tree. The red solid dot is the node position and the position of the red square is the grafting position of the species.

**Table 1 entropy-21-00313-t001:** The number of possible values.

No. Attributes	No. Values	Possible Value
1	2	$\left\{0\right\}\left\{1\right\}$
2	$3^{2}$	$\left\{0,0\right\}\left\{0,1\right\}\left\{0,2\right\}\left\{1,0\right\}\left\{1,1\right\}\left\{1,2\right\}\left\{2,0\right\}\left\{2,1\right\}\left\{2,2\right\}$
3	$4^{3}$	$\left\{0,0,0\right\}\left\{0,0,1\right\}\left\{0,0,2\right\}\left\{0,0,3\right\}\left\{0,1,0\right\}\left\{0,1,1\right\}\left\{0,1,2\right\}\left\{0,1,3\right\}$
…	…	…
*N*	${\left(N+1\right)}^{N}$	$\left\{0,0,\ldots ,0\right\}\left\{0,0,\ldots ,1\right\}\left\{0,0,\ldots ,2\right\}\ldots \left\{0,0,\ldots ,N\right\}\left\{0,1,\ldots ,0\right\}$

**Table 2 entropy-21-00313-t002:** Setting the chromosome bit and code.

Site	1	2	3	4	5	6	7	8	9	10
Code	1	3	0	4	5	7	10	9	9	8

**Table 3 entropy-21-00313-t003:** Experimental data sets.

Datasets	No. Species	No. Attributes	Reference
Pharyngodonidae	25	30	Bouamer and Morand (2003) [35]
Hibiscus	40	38	Tang et al. (2014) [36]
Meligethes	42	60	Lin et al. (2015) [37]
Nemesiid spiders	77	60	Goloboff (1995) [38]
Phrynosomatid lizards	115	59	Reeder and Wiens (1996) [39]
liebherr	160	136	Hawaiian Platynini (Carabidae), Liebherr (1998) [40]

**Table 4 entropy-21-00313-t004:** Average accuracies of the different methods for different data sets. The bold numbers indicate the highest accuracy in the column.

	Pharyngodonidae	*Hibiscus*	*Meligethes*	Nemesiid Spiders	Phrynosomatid Lizards	*Liebherr*	Avg.
BI	0.8919	0.8714	0.7828	0.8672	0.8567	0.8355	0.851
ML	0.8400	0.8250	0.7461	0.8659	0.8501	0.8428	0.828
MP	**0.8990**	0.8778	0.7905	0.8730	0.8618	0.8283	0.855
CDT	0.8983	**0.8811**	**0.7930**	**0.8811**	**0.8732**	**0.8613**	**0.865**

**Table 5 entropy-21-00313-t005:** The variance of tree length between the CDT algorithm and that calculated by the other three methods.

	Pharyngodonidae	*Hibiscus*	*Meligethes*	Nemesiid Spiders	Phrynosomatid Lizards	*Liebherr*	Avg.
CDT vs. BI	0.0282	0.0800	0.4278	0.0282	0.0800	0.0500	0.1157
CDT vs. ML	0.0153	0.0957	0.0488	0.0153	0.0957	0.0180	0.0481
CDT vs. MP	0.1250	0.1128	0.0282	0.1250	0.1128	0.0821	0.0977

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
