# Peer review of "A New Phylogenetic Inference Based on Genetic Attribute Reduction for Morphological Data"

_entropy, 2019, doi:10.3390/e21030313_

Round 1
Reviewer 1 Report
The authors have investigated the bi-directional cognitive and concept-driven processing for the process of phylogenetic inference. This study solves the problem of building a stable phylogenetic tree when much data is missing. This paper will have several applications in genetics and related fields. Given that the algorithm works out for many missing data, they could be replaced by zero elements in a matrix approach, that is a sparse matrix. As a result, the bibliography should present some link the wavelet analysis. For this reason, I suggest adding the references given below.
1) Mallat S.. A Theory for Multiresolution Signal Decomposition: The Wavelet Representation. IEEE Transactions on Pattern Recognition and Machine Intelligence 1989, 11 (7), 674-693.
2) Guido R.C.; Addison p.; Walker J. Introducing wavelets and time-frequency analysis, IEEE Eng. Biol. Med. Mag. 2009, 28(5), 13 .
3) Guariglia E.; Silvestrov S. (2017). Fractional-Wavelet Analysis of Positive definite
Distributions and Wavelets on D’(C), in Engineering Mathematics II, Silvestrov, Rancic
(Eds.), Springer, pp. 337-353.
4) Daubechies I. Ten Lectures on Wavelets. SIAM, Philadelphia (1992).
5) Newland D.E. Harmonic Wavelet Analysis. Proceedings of The Royal Society A 1993, 443(1917), 203-225.
6) Guariglia E. Spectral Analysis of the Weierstrass-Mandelbrot Function. IEEE Conference Proceedings 2017. In: Proceeding of the 2nd International Multidisciplinary Conference on Computer and Energy Science, Split, Croatia, 12-14 July 2017.
Moreover, I strongly recommend an additional English review.
Reviewer 2 Report
The authors developed an interesting paper to address the instability of phylogenetic trees in morphological datasets. Specifically, they present a phylogenetic inference method based on a Concept Decision Tree (CDT).Obtained results indicate that as the proportion of missing data in the phylogenetic analyses increases, CDT can remain 86.5% accurate and produce a reliable phylogenetic tree.
I consider that the paper presents novelty and an update for the literature. However, I confess that my expertise is related to information theory and not in genetic inference; so I attached below some comments to help to authors in the review of the manuscript before accept the recommendation:
Moderate comments:
Eq. (5): Here, given that mutual information is a commonly tool used by several researchers, I recommends to add a reference: Arellano-Valle et al. (2013). Also, the author could includes a property of MI: Let $X$ and $Y$ be two random variables, then, $I(X,Y)=0$ when $X$ and $Y$ are independent; otherwise, this index is positive (Arellano-Valle et al., 2013; Cover & Thomas, 2006), and it increases with the degree of dependence between the components $x_i$ and $y_i$.
L267-268: Please, provide more details about the use of these three methods.
Figure 7: Please, increase the size of this figure to view your results in a better form.
Minor comments:
L178: "defines" <-> "is defined as".
L180: ". If".
L200: "Reduces (".
L247: Given that a_8 is missing data, delete "a_8=?".
References:
Arellano-Valle, R.B., Contreras-Reyes, J. E., Genton, M. G. (2013). Shannon Entropy and Mutual Information for Multivariate Skew‐Elliptical Distributions. Scandinavian Journal of Statistics, 40(1), 42-62.
Cover, T.M., Thomas, A.J. (2006). Elements of information theory. 2nd edition. Willey-Interscience, NJ, USA.
Author Response
Dear professor: Thank you for your comments on our manuscript “A New Phylogenetic Inference Based on Genetic Attribute Reduction for Morphological Data” (entropy-451644). Based on these comments and suggestions, we have made careful modifications on the original manuscript. Below you will find our point-by-point responses to the comments/questions. I am looking forward to hearing from you. Thank you and best regards. Yours sincerely, Zeyun Liu E-mail: liuzeyun@stumail.nwu.edu.cn
Round 2
Reviewer 1 Report
The requested revision has not been applied. The English is still weak as well as the bibliography.
Author Response
Dear professor:
Thank you for your comments on our manuscript "A New Phylogenetic Inference Based on Genetic Attribute Reduction for Morphological Data"(entropy-451644).
Based on these comments and suggestions, we have made careful modifications on the manuscript. Below you will find our point-by-point responses to the comments. I am looking forward to hearing from you.
Thank you and best regards.
Yours sincerely,
Zeyun Liu
E-mail: liuzeyun@stumail.nwu.edu.cn

Round 3
Reviewer 1 Report
The requested revision has been applied.